# Assessing the impact of a tennis-themed sports video game on physical literacy and participation on physical activities among university students: A cluster randomized controlled trial study protocol

Wai Keung Ho[1,2], Raymond Kim Wai Sum[1]*, Siu Ming Choi[3]

**1** Department of Sports Science and Physical Education, Faculty of Education, The Chinese University of Hong Kong, Hong Kong, China, **2** Physical Education Unit, Faculty of Education, The Chinese University of Hong Kong, Hong Kong, China, **3** Faculty of Education, The University of Macau, Macau, China

\* kwsum@cuhk.edu.hk

## Abstract

### Introduction

Physical literacy (PL) is a multidimensional construct encompassing motivation, confidence, physical competence, knowledge, and understanding, which is essential for fostering lifelong physical activity. University students represent a critical stage for PL development, yet many experience a decline in physical activity and an increase in sedentary behavior after high school. Sports video games (SVGs), though traditionally sedentary, have shown potential to enhance motivation and cognitive engagement with sports. These games are especially popular among university students, making them an accessible and engaging tool for promoting PL. Despite their popularity, the role of SVGs in supporting PL development remains underexplored, particularly in Asian contexts. Integrating technology into physical education, such as through SVGs and virtual simulations, may offer innovative ways to enhance engagement and support PL development. Although the intervention is sedentary, SVG gameplay often requires reflexes and eye–hand coordination which may influence aspects of physical competence such as reaction time and coordination. This study aims to evaluate whether a tennis-themed SVG intervention integrated into PE lessons can enhance students' motivation, sport-specific knowledge, physical competence, social engagement, and physical activity engagement, thereby promoting holistic PL development.

### Methods

A cluster randomized controlled trial will be conducted with 176 first-year students enrolled in compulsory tennis physical education courses at The Chinese University

**Data availability statement:** No datasets were generated or analysed for this manuscript because it describes a study protocol.

**Funding:** The author(s) received no specific funding for this work.

**Competing interests:** The authors have declared that no competing interests exist.

**Abbreviations:** AHWT: Alternate Hand Wall Toss test; ANOVA: Analysis of Variance; APLQ: Adolescent Physical Literacy Questionnaire; CUHK: The Chinese University of Hong Kong; ICC: Intra-cluster Correlation; IPAQ-SF: International Physical Activity Questionnaire – Short Form; MANCOVA: Multivariate Analyss of Covariance; OSF: Open Science Framework; PE: Physical education; PI: Principal investigator; PL: Physical literacy; PPLI: Perceived Physical Literacy Instrument; RDT: Ruler Drop Test; SMD: Standardized mean difference; SMS-6: Sport Motivation Scale-6; SPIRIT: Standard Protocol Items: Recommendations for Interventional Trials; SVG: Sports video game.

of Hong Kong. Eight entire classes (four male and four female) will be randomly assigned to either an intervention or control group. The intervention group will play a tennis SVG, *TopSpin 2K25,* twice weekly for six weeks alongside regular PE lessons. Assessments at baseline, post-intervention, and three-week follow-up will include self-reported gaming behavior, physical activity level, PL domains, and objective tests of reaction time, eye-hand coordination, and tennis knowledge.

## Discussion

This protocol examines the integration of SVG into physical education to promote PL across physical, cognitive, affective, and social domains. Limitations include potential contamination bias, variability in prior gaming experience, short intervention duration, and reliance on self-reported measures, which may introduce bias. Theoretically, the study advances PL frameworks by embedding technology-enhanced learning within structured PE contexts. Practically, SVG-based activities offer an inclusive, engaging approach to support diverse learners. Future research should explore long-term effects and develop validated tools for comprehensive assessment.

**Trial registration:** The protocol has been registered on the Open Science Framework on 25 July 2025: https://doi.org/10.17605/OSF.IO/JDYGV.

## Introduction

Physical literacy (PL) is increasingly recognized as a foundational concept in physical education, encompassing the motivation, confidence, physical competence, knowledge, and understanding necessary for individuals to value and engage in physical activity throughout life [1]. For university students, who are transitioning into adulthood and independence, developing PL is particularly critical. This stage is often considered the final opportunity within formal education to instill essential PL attributes and foster a lifelong positive attitude toward physical activity [2]. However, after graduating from high school, many adolescents show a decline in physical activity participation [3], alongside a rise in sedentary leisure behaviors [4]. These trends present significant challenges to the development of sustainable physical activity habits and comprehensive PL. While PL is widely accepted as a guiding framework, ongoing theoretical debates persist regarding its measurement and operationalization. For example, Boldovskaia et al. [5] emphasize that despite PL's conceptual acceptance, there is considerable debate over appropriate methods for collecting empirical data, with most existing instruments focusing narrowly on certain dimensions.

Traditional physical education (PE) programs have primarily focused on developing motor skills and fitness, often overlooking opportunities to integrate technology-driven approaches that can enhance engagement and learning. Recent advancements in educational technology have opened new pathways for promoting PL through interactive and immersive experiences. AI-assisted interactive systems have demonstrated increased engagement and comprehension in

technology-mediated environments, supporting the educational potential of game-based interventions [6]. These systems leverage adaptive feedback and personalized learning to create dynamic environments that foster motivation and skill acquisition. In the context of PE, sports video gaming (SVG) represents a promising application of these principles, offering students an engaging platform to practice decision-making, strategy, and collaboration while reinforcing physical skills.

Beyond PL, previous research indicates that structured physical activity interventions can yield significant psychological and physiological benefits, including mood regulation, emotional resilience, and neurochemical modulation [7]. These findings underscore the broader role of physical engagement through innovative modalities such as SVGs to promote PL, enhance motivation, and support overall health among university students.

At the same time, video gaming has become the most popular form of entertainment among adolescents and young adults worldwide. Traditional video games are typically played in a stationary posture using simple hand or finger movements, and are therefore considered a form of sedentary behavior [8]. Among these, SVGs are especially popular among adolescents. In 2024, five SVGs ranked among the top ten best-selling titles in the United States, including *EA Sports College Football 25*, *NBA 2K25* and *Madden NFL 25* [9]. SVGs simulate traditional sports using handheld controllers to manipulate on-screen avatars in a seated position [10]. Despite their sedentary nature, research has shown that SVGs can increase motivation and interest in participating in real-life sports among youth and adolescents [11–14]. According to Adachi and Willoughby [11], playing SVGs can foster a sense of competence and mastery, which boosts self-confidence and encourages involvement in physical sports. SVGs also simulate sports environments and strategies, increasing cognitive familiarity and reducing barriers to real-world participation. Additionally, studies suggest that SVGs can be effective tools for teaching sports-related knowledge, such as rules, strategies, and tactics especially for beginners [13,15,16].

While existing research has largely focused on the affective and cognitive benefits of SVGs, there remains a notable gap in studies exploring their influence on physical competence and the social domains of PL [17]. Crucially, the PL construct consists of four interrelated elements (motivation/confidence, physical competence, knowledge/understanding, engagement in physical activities for life) and respective domains (affective, physical, cognitive, behavioral) [18]. Improvements in one domain can positively influence others; for example, greater motivation may increase activity participation, enhancing physical competence and confidence, while deeper sport-specific knowledge can foster meaningful social interactions. SVGs, by stimulating motivation and cognitive engagement, may serve as a catalyst for holistic PL development. Moreover, gameplay often involves perceptual-motor demands such as reflexes and coordination and incorporates cooperative or competitive social interactions. These considerations underscore the need for interventions addressing all PL domains rather than focusing narrowly on motivation or knowledge.

In this study, simple reaction time and eye–hand coordination were selected as outcome measures because they represent targeted components within the physical domain of PL and align with the perceptual–motor demands of both video gaming and real-life tennis. While these measures do not capture comprehensive motor competence, they provide insight into specific physical attributes that may be influenced by the intervention.

A tennis-themed SVG intervention was selected for this study based on the limited exposure to tennis among university students in Hong Kong. Tennis is not commonly taught in primary or secondary education, resulting in most students entering university with minimal prior experience in the sport. Teaching tennis in PE classes for beginners presents several challenges. Due to time constraints and varying skill levels, it is difficult to cover all fundamental techniques and tactical concepts, and students may struggle to apply what they learn in real game situations. The selected SVG (TopSpin 2K25) is a well-developed and immersive tennis simulation that addresses these limitations by providing repeated exposure to tennis rules, scoring systems, basic techniques, and gameplay strategies in a visually engaging and easy-to-understand format. This allows students to build sport-specific knowledge and decision-making skills at their own pace, complementing the learning objectives of the PE curriculum.

The intervention leverages specific game mechanics to target PL domains and foster motivation for real-life tennis participation rather than increased video gaming. Fast-paced rally sequences and timing-based challenges aim to improve

perceptual-motor skills such as reaction time and eye-hand coordination (physical domain). In-game tutorials and rule-based decision-making tasks support sport-specific knowledge (cognitive domain). Immersive graphics, realistic physics, and authentic match scenarios are designed to enhance enjoyment and create a sense of competence, which may translate into motivation to engage in actual tennis (affective domain). Multiplayer modes encourage peer interaction and collaboration (behavioral domain). Together, these mechanics are intended to promote multidimensional PL development and bridge virtual engagement with real-world physical activity.

The intervention is designed to be flexible, allowing it to enhance rather than replace traditional PE instruction. Since the SVG does not require formal teaching and students engage with the game independently, it can be integrated as a supplementary activity outside of scheduled PE class time. This approach enables students to reinforce and extend their learning of tennis-related concepts without interfering with the core instructional time allocated for physical skill development. The SVG provides additional opportunities for cognitive engagement and sport-specific knowledge acquisition, supporting the broader goals of PL.

Despite growing interest in the educational and motivational potential of SVGs, intervention studies utilizing these tools to promote PL remain limited, particularly in Asian contexts [17]. Most existing research has been conducted in Western settings, leaving a gap in understanding how culturally relevant applications of SVGs may influence PL and physical activity behaviors in Asian populations.

Therefore, the aim of this study is to investigate the effectiveness of a tennis-themed SVG intervention, implemented alongside regular PE lessons, in promoting holistic PL among university students in Hong Kong. Specifically, the study will evaluate whether SVG participation can enhance motivation, sport-specific knowledge, physical competence, social engagement, and physical activity engagement. Using a cluster randomized controlled trial design, this research seeks to identify innovative, technology-enhanced strategies to improve PE outcomes and foster lifelong physical activity.

Following the intervention and subsequent follow-up, we hypothesize that:

1. *Physical Competence*: Participants in the intervention group will demonstrate greater improvement in perceptual-motor skills, specifically simple reaction time and eye-hand coordination, compared to those in the control group. These measures represent targeted components within the physical domain of PL and align with the perceptual-motor demands of both sports video gameplay and real-life tennis, rather than comprehensive motor competence.

2. *Motivation for real-life sports*: Participants in the intervention group will report a greater increase in motivation to engage in real-life sports compared to those in the control group.

3. *Cognitive knowledge of tennis:* Participants in the intervention group will achieve a greater improvement in cognitive knowledge related to tennis than participants in the control group.

4. *Social engagement in physical activities*: Participants in the intervention group will exhibit greater enhancement in in-person social interaction relative to the control group.

5. *Physical activity engagement:* Participants in the intervention group will show a higher increase in physical activity engagement levels than their counterparts in the control group.

## Materials and methods

### Study design

A cluster randomized controlled trial will be conducted to examine the effectiveness of participation in the tennis SVG '*TopSpin 2K25*' on PL among university students at the Chinese University of Hong Kong (CUHK), beyond their regular tennis PE lessons. In this context, the PE courses refer specifically to the public PE practice courses that are compulsory for non-sports majors, guided by PE teachers, which are credit-bearing. Our study employs cluster randomization at the

class level, rather than individual randomization, which is commonly used in educational setting. This approach allows entire classes to be randomly assigned to either the intervention or control group, maintaining the integrity of existing class structures and minimizing disruption to timetables and teaching logistics.

A total of 176 first-year university students enrolled in tennis PE courses will be recruited. These students will be drawn from eight entire classes – four male and four female, with each class comprising 22 students. Although students self-select their PE classes during course registration, random assignment will be conducted at the class level after enrollment is finalized. The classes will be randomly assigned to either the intervention group (two male and two female classes; 88 students) or the control group (two male and two female classes; 88 students).

The intervention group will engage in two 20-min sports video gaming sessions weekly in addition to their regular PE lessons. The intervention will last for six weeks, followed by a three-week follow-up period. Data will be collected at three time points: baseline, post-intervention, and follow-up. The intervention duration of two 20-minute sessions per week was selected based on feasibility within the university schedule and supported by prior research. Röglin et al. [19] conducted a randomized controlled trial in which 15–20-minute exergaming sessions, delivered twice weekly over three months, significantly improved students' physical self-concept, including strength, coordination, and speed. These findings suggest that even short-duration, structured gaming interventions can elicit measurable changes in physical competence and support physical literacy development.

The study design, implementation and reporting of the results will adhere to the Standard Protocol Items: Recommendations for Interventional Trials (SPIRIT) guidelines [20]. This study protocol was prospectively registered on the Open Science Framework (OSF) on 25 July, 2025. The full protocol is accessible at: https://doi.org/10.17605/OSF.IO/JDYGV.

### Participants

**Sample size calculation.** Given that the hypotheses in this study involve both between-subjects comparisons (intervention vs. control groups) and within-subjects measurements across three time points, the sample size was calculated using the *"ANOVA: Repeated measures, within-between interaction"* option in G*Power 3.1 software. Based on an expected medium effect size ($f = 0.25$), an alpha level of 0.05, and a desired statistical power of 0.80, the required sample size was determined to be 28 participants per group. The choice of $f = 0.25$ was informed by Cohen's conventional benchmarks and supported by findings from related educational and exergaming interventions. For instance, Zhao et al. [21] conducted a meta-analysis of 16 randomized controlled trials involving 2,962 students and found that exergames significantly improved PE learning outcomes, with a standardized mean difference (SMD) of 0.45, which corresponds to a medium effect size. Given the educational nature of our intervention and the exploratory context, this estimate was considered both conservative and realistic. To account for an anticipated dropout rate of 20%, the target sample size was increased to 35 participants per group, resulting in a minimum total of 70 participants. To ensure sufficient power and account for unforeseen attrition, more participants than the minimum required will be recruited.

Since this study employs a cluster randomized controlled trial design, it is essential to account for the intra-cluster correlation (ICC) to avoid underestimating the required sample size. Participants within the same class may exhibit correlated outcomes due to shared learning environments and peer interactions. To adjust for clustering, we applied the design effect formula:

$$\text{Design Effect} = 1 + (m - 1) \times ICC$$

Where:

• $m$ = average cluster size (22 students per class)

• $ICC$ = assumed intra-cluster correlation (conservatively estimated at **0.02** based on similar educational interventions)

Thus:

$$\text{Design Effect} = 1 + (22 - 1) \times 0.02 = 1 + 21 \times 0.02 = 1.42$$

The adjusted sample size becomes:

$$70 \times 1.42 \approx 100 \text{ participants}$$

To ensure adequate power and account for potential attrition and clustering effects, we plan to recruit 176 participants across 8 classes, which exceeds the adjusted minimum requirement.

### Eligibility criteria and recruitment

Eligible participants must be first-year students enrolled in the required tennis PE course, aged 18 or above, and free from any disabilities, injuries, or illnesses that would impair their ability to perform daily tasks. Students with a regular habit of playing *TopSpin 2K25* will be excluded.

This study is scheduled to begin in the first week of the second semester of the 2025–2026 academic year in CUHK. A briefing session and participant recruitment will be conducted during the week of 5 January 2026, integrated into the schedule tennis PE lessons. Recruitment will conclude on 16 January 2026, coinciding with the end of the university's add-drop period, at which point the class lists will be finalized. Recruitment will take place during the first lesson in the PE tennis courses, where teachers will explain the study and distribute consent forms. An introductory meeting will be held with the teachers to outline the intervention procedures and assessment methods. Additionally, a training workshop will be provided for research assistants and student helpers, covering gameplay instructions and guidelines for facilitating the gaming sessions.

### Randomization

Each semester, CUHK offers seven male and seven female tennis PE classes, all taught by different teachers but following standardized content and assessments. For this study, simple randomization will be used. Four male and four female entire classes will be randomly selected by drawing lots. Among these, two male and two female entire classes will be further randomly assigned to the intervention group using the same method.

### Intervention

A six-week tennis-themed SVG intervention will be implemented for participants in the intervention group, supplementing their regular tennis PE lessons. This intervention is designed to align with the study's goals of promoting PL and enhancing physical activity engagement among university students in Hong Kong.

A dedicated **"SVG Corner"** will be set up in a classroom, equipped with two PlayStation 5 consoles, each connected to four controllers and a monitor. Participants will engage with the tennis-themed SVG '*TopSpin 2K25*', released in April 2024, which is known for its highly realistic gameplay, including accurate physics, player movement, and court dynamics. This realism helps bridge the gap between virtual and physical tennis, making it a suitable tool for reinforcing PL concepts such as spatial awareness, timing, and movement strategies. The game includes tutorials and skill challenges that provide structured learning opportunities aligned with specific PL goals. For example:

1. **Physical Competence**:

Gameplay requires fast reaction time and eye-hand coordination, simulating real tennis movements and decision-making. These elements promote motor skill development, agility, and movement control, which are core aspects of physical competence.

 

2. **Motivation**:

The immersive and gamified nature of *TopSpin 2K25* enhances enjoyment and engagement. By offering a fun and competitive environment, the intervention aims to boost students' overall interest and motivation in real-life sports, supporting long-term physical activity participation.

3. **Knowledge and Understanding**:

In-game tutorials and challenges help students understand tennis rules, tactics, and movement principles, reinforcing cognitive engagement with physical activity.

4. **Social Interaction**:

Multiplayer modes (singles and doubles) promote peer interaction, cooperation, and friendly competition, which are important social dimensions of PL.

To support accessibility and adherence, at least one research assistant or student helper will be present during operating hours to manage the SVG Corner and assist participants. Clear and concise control guidelines will be displayed in the SVG Corner for easy reference. To maintain the integrity of the intervention, no formal tennis instruction will be provided during gaming sessions.

Each participant in the intervention group will be required to attend two 20-minute gaming sessions per week, in addition to their regular PE classes. The SVG Corner will be open Monday to Friday, from 10:00–13:00 and 14:00–18:00. Due to equipment limitations (2 consoles with a total of 8 controllers), each session has limited quota. If a session reaches its quota due to the first-come-first-served arrangement, we will arrange an alternative time slot to accommodate affected participants. The attendance will be tracked weekly, in case where participants are unable to fulfill the attendance requirement, we will contact them via email to arrange make-up sessions. This ensures that all participants can fulfill the attendance requirement and receive the full intended dosage of the intervention.

To minimize contamination bias, students in the control group will not be provided access to the SVG Corner and will be advised not to engage in tennis-themed sports video gaming independently during the study period. While we cannot fully restrict external gaming behavior, participants will be informed of the importance of maintaining their assigned condition and asked to report any SVG-related activity outside the study. This will help us monitor potential crossover and account for it in the final analysis.

The first two sessions will include guided tutorials within the game to help participants become familiar with the controller layout and basic tennis concepts. For example, the initial tutorial focuses on repositioning, requiring players to return to the center area behind the baseline after each shot. An instructional video will explain the importance of repositioning and demonstrate how to control the avatar. Participants must successfully complete six repositioning tasks to move forward. From the third session onward, participants will engage in both singles and doubles matches, playing with or against other participants. No gaming sessions will be provided to the control group. Participants may withdraw from the intervention if they experience scheduling conflicts, physical discomfort, musculoskeletal or cardiovascular conditions, or are unable to comply with the intervention protocol. To support adherence, PE teachers and research assistants will regularly remind participants to attend their gaming sessions. Additionally, all participants in the intervention and control groups will receive four participation marks in their PE course upon completing the study.

## Data collection

This study will employ both self-reported and objective measures to comprehensively assess participants' PL, engagement in physical activities, and sports video gaming habits.

Self-reported data will be collected through a structured set of online questionnaires (see Appendix 1), which include items on video gaming behavior, physical activity levels, physical literacy, and demographic information. These

questionnaires will be administered by trained research assistants using Qualtrics, a secure and institution-approved online survey platform. Access to the survey will be provided via a secure link or QR code, distributed to students by their teachers.

Objective measures will include assessments of simple reaction time and eye-hand coordination, conducted using the Ruler Drop Test (RDT) and the Alternate Hand Wall Toss Test (AHWT), respectively. Additionally, a knowledge-based quiz will be administered to evaluate participants' cognitive understanding of tennis (see Appendix 2).

All assessments will be conducted during scheduled PE tennis lessons at the CUHK. Data collection will occur at three time points: baseline, post-intervention, and a three-week follow-up. Baseline data collection is scheduled for the first week of the second semester (early January 2026), during the same lesson in which participants are recruited. This will be followed by a six-week sports video gaming (SVG) intervention. Post-intervention assessments will be conducted in Week 8 (early March 2026), and a three-week follow-up assessment will take place in Week 11 (late March 2026).

The participant timeline is illustrated in Fig 1. To ensure consistency across all groups, identical procedures and instruments will be used. All assessments will be administered by the principal researcher in collaboration with trained research assistants. A summary of the measurement tools and outcome variables is provided in Table 1.

1. **Demographic information**

Participants will provide demographic details including age, gender, ethnicity, academic department and faculty, sports regularly participated in, and years of involvement in those sports. The details will be collected in the beginning of the online questionnaire.

2. **Self-reported measures**

To capture participants' behaviors and perceptions across relevant domains, a series of self-reported instruments will be administered via an online questionnaire.

i) *Video gaming habit*
Participants will respond to questions regarding their video gaming behaviors, including frequency, duration, devices used, and types of games played. Sample items include:

• "How much time per week you play video games?"

• "How many days per week do you play video games?"

• "How long have you been playing video games?"

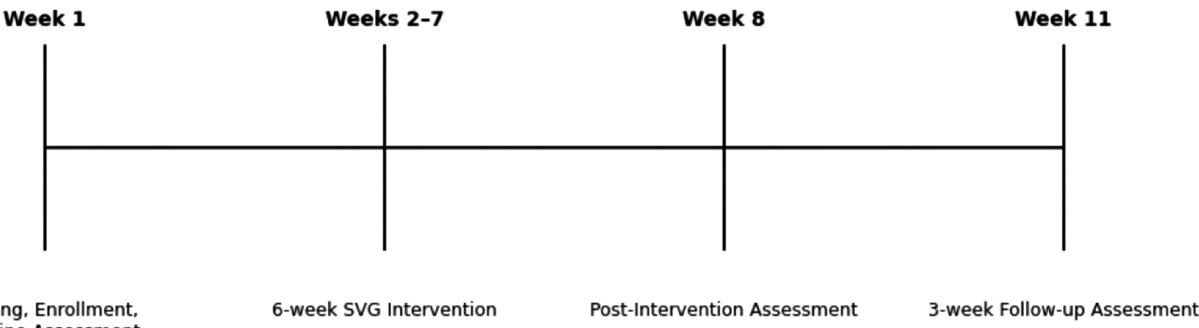

**Fig 1. Participant timeline schematic diagram.** Week 1 included briefing, enrollment, and baseline assessment. Weeks 2-7 comprised the 6-week SVG intervention. Post-intervention assessment was conducted in Week 8, followed by a 3-week follow-up assessment in Week 11.

**Table 1. Summary of measurements in this study.**

| Instrument/Method used | Outcome assessed | Type | Physical literacy domain | Reference |
|---|---|---|---|---|
| Custom questionnaire on video gaming | Video gaming habits and demographics | Self-reported | Not applicable | Developed by researcher |
| International Physical Activity Questionnaire – Short Form (IPAQ-SF) | Physical activity level | Self-reported | Physical | [22] |
| Perceived Physical Literacy Instrument (PPLI) | Perceived physical literacy | Self-reported | Affective, social, cognitive | [23] |
| Sport Motivation Scale-6 (SMS-6) | Motivation for sport participation | Self-reported | Affective | [25] |
| Selected items from Adolescent Physical Literacy Questionnaire (APLQ) | Social engagement in physical activity | Self-reported | Social | [24] |
| Ruler Drop Test (RDT) | Simple reaction time | Objective | Physical | [26] |
| Alternate Hand Wall Toss Test (AHWT) | Eye-hand coordination | Objective | Physical | [27] |
| Researcher-designed multiple-choice quiz | Knowledge of tennis | Objective | Cognitive | Developed by researcher |

Participants who indicate regular gaming activity will be asked additional questions specific to sports video gaming (SVG), such as:

• "Which types of sports video games you play?"

• "How much time per week you play sports video games?"

• "How many days per week do you play sports video games?"

ii)  *Physical activity level*

Participants' physical activity levels will be assessed using the International Physical Activity Questionnaire – Short Form (IPAQ-SF)**.** This instrument evaluates the frequency and duration of physical activities across different intensities: vigorous, moderate, and walking, as well as time spent sitting, over the past seven days. It provides an estimate of total physical activity expressed in MET-minutes per week. A sample item includes:

• "During the last 7 days, on how many days did you do vigorous physical activities like heavy lifting, digging, aerobics, or fast bicycling?"

The IPAQ-SF is suitable for use with individuals aged 15 years and older. Its reliability and validity have been supported by a large-scale international study conducted by Craig et al. [22], which involved participants from 12 countries. The study reported acceptable test-retest reliability, with correlations typically exceeding 0.80, and moderate criterion validity, with correlations ranging from 0.30 to 0.50 when compared to accelerometer data.

iii)  *Perceived physical literacy*

To assess PL comprehensively, we adopted a multi-instrument approach. Perceived PL will be measured using the Perceived Physical Literacy Instrument (PPLI), a 9-item scale comprising three subscales: *knowledge and understanding*, *self-expression and communication with others*, and *sense of self and self-confidence*. Responses are recorded on a 5-point Likert scale. The instrument has demonstrated satisfactory construct validity (RMSEA = 0.08; CFI = 0.94; SRMR = 0.04) and convergent validity (CR = 0.72–0.78; AVE = 0.43–0.54) in adolescent populations [23].

iv)  *Social engagement in physical activities*

To assess the social dimension of PL, four items from the Adolescent Physical Literacy Questionnaire (APLQ) will be employed. The APLQ is a validated 25-item instrument designed to measure various domains of physical literacy among

adolescents [24]. It was selected because it is the only available validated tool that explicitly includes the social domain, which was central to this study's objectives. For the purposes of this study, four items specifically targeting social engagement in physical activities have been selected. Participants will respond to each item using a 5-point Likert scale ranging from "strongly disagree" to "strongly agree."

The selected items are:

• "I do sports and physical activity with family or friends."

• "I encourage others to do sports and physical activity with me."

• "I participate in group sports outside of school time."

• "I have made new friendships in sport and physical activity."

These items are intended to capture participants' interpersonal experiences and social involvement in physical activity, which are recognized as key components of PL development.

v)      *Motivation in engagement in sports*

Motivation for engaging in sports will be assessed using the Sport Motivation Scale-6 (SMS-6), a 24-item instrument designed to measure contextual motivation in sport. The scale includes six subscales representing different types of intrinsic and extrinsic motivation. Responses are recorded on a 7-point Likert scale. The SMS-6 has demonstrated satisfactory construct validity in athletic populations [25].

3.  **Objective measures**

To complement the self-reported data, three objective assessments will be conducted to evaluate participants' physical and cognitive performance: simple reaction time, eye-hand coordination, and cognitive knowledge in tennis.

i)      *Simple reaction time*

Simple reaction time will be measured using the Ruler Drop Test (RDT). A 50-cm ruler will be used for the assessment. Participants will be seated with their arm in a mid-prone position and elbow flexed at 90°. The ruler will be held vertically with the zero-mark aligned at the participant's fingertips. Upon release, participants will attempt to catch the ruler as quickly as possible. The distance the ruler falls before being caught will be recorded in centimeters. Each participant will complete three trials with each hand, and the average of the three measurements will be used for analysis. Prior to testing, participants will be given several practices attempts to familiarize themselves with the procedure. Reaction time will be calculated based on the distance fallen, using the standard formula for free fall under gravity [26].

ii)  *Eye-hand coordination*

Eye-hand coordination will be assessed using the Alternate Hand Wall Toss Test (AHWT), a widely used measure in sports medicine [27]. Participants will stand 2.0 meters from a wall and throw a tennis ball underhand with one hand, attempting to catch it with the opposite hand. The sequence will alternate between right-hand throws with left-hand catches and vice versa. The total number of successful catches within 30 seconds will be recorded. Each participant will complete two trials, and the best score will be used for analysis. Practice attempts will be allowed before the formal assessment.

iii)  *Cognitive knowledge in tennis*

Cognitive understanding of tennis will be evaluated using a researcher-designed multiple-choice quiz consisting of ten items, each with four answer options (see Appendix 2). The quiz covers essential content areas including tennis rules, terminology, court layout, tactics, and skill-related knowledge. Items were developed collaboratively by PE teachers with expertise in tennis, drawing from both the instructional content of the tennis PE course and the tennis-related elements embedded in the SVG.

The quiz was pilot-tested with a small group of university students ($n = 10$) to evaluate clarity and appropriateness for the target population. Feedback from the pilot group informed refinements to the questions, supporting the content validity of the instrument. Although the quiz has not undergone formal psychometric validation, it is tailored to reflect the specific learning objectives of the intervention. To minimize response bias, the order of answer choices will be randomized across different assessment time points. Sample questions include:

- "What is the term for a serve that the opponent cannot touch?"
- "What happens if a player serves two consecutive faults?"
- "Which strategy involves hitting the ball deep into the opponent's court to push them back?"
- "What is the primary goal of the "one up, one back" formation in doubles?"

## Ethics approval and consent to participate

Ethical approval for this study has been obtained from the Survey and Behavioural Research Ethics Committee of CUHK (Ref. No.: SBRE-24–0047). All procedures involving human participants will be conducted in accordance with the ethical standards of the institutional and/or nation research committee and with the 2024 revision of the Declaration of Helsinki (https://www.wma.net/policies-post/wma-declaration-of-helsinki/). Written informed consent will be obtained from all participants prior to their involvement in the study.

## Dissemination plan

Findings from the study will be disseminated through peer-reviewed journal publications, presentations at international conferences, and academic lectures. For requests related to secondary data analysis, interested parties should contact the corresponding author for permission.

## Data analysis

Descriptive statistics, including means and standard deviations, will be calculated for university students' video gaming time, gaming frequency, physical activity levels, physical literacy, reaction time, and eye-hand coordination across all relevant subscales and instruments. To examine baseline differences between groups in the cross-sectional study, one-way analyses of variance (ANOVAs) will be conducted.

To evaluate intervention effects, mixed-effects models will be employed instead of simple repeated-measures ANOVAs to account for the hierarchical structure of the data and potential intra-cluster correlation among participants within the same class group. This approach addresses the reality that, although courses follow standardized content, variability in teaching style, instructor enthusiasm, and informal reinforcement of the intervention is inevitable in real-world PE settings. Mixed-effects models will include fixed effects for time (baseline, post-intervention, follow-up) and group (intervention vs. control), as well as random effects for class clusters to capture variability across different teaching contexts. These models will assess both main effects and interaction effects, providing more accurate standard error estimation and valid statistical inference.

For secondary outcomes (motivation, cognitive knowledge, social engagement, and physical activity), similar mixed-effects or generalized estimating equation models will be applied, depending on distributional assumptions. Multiple linear regression analyses will explore predictive relationships between PL and physical activity levels, with SVG gaming included as a potential mediator or moderator. Covariates such as baseline physical activity, prior SVG experience, and demographic characteristics (age, gender) will be adjusted for in all models. Interaction effects between the intervention and participant characteristics (e.g., gender, gaming experience) will also be examined to identify subgroup differences.

To enhance the interpretability and transparency of our findings, partial eta-squared will be reported for all primary outcomes to indicate the magnitude of observed effects. Additionally, 95% confidence intervals will be provided to reflect the precision of the estimates and the range within which the true effect is likely to fall. These metrics will support a more nuanced understanding of the intervention's impact and facilitate comparison with other studies.

To ensure the integrity of the analysis, the following procedures will be used to address missing data. Participants with more than 20% missing responses on key outcome measures (e.g., physical literacy or physical activity questionnaires) may be excluded from specific analyses. All decisions regarding missing data handling will be documented in the final report, including the rationale for exclusion or imputation. All statistical analyses will be performed using SPSS Version 30 for Windows, with the significance level set at $p < 0.05$.

## Discussion

### Limitations

This study protocol has several limitations that should be acknowledged. First, although efforts will be made to minimize contamination bias, it is not possible to fully restrict students in the control group from engaging in tennis-themed sports video gaming independently outside the study setting. Such exposure may introduce variability and potentially dilute the observed intervention effects. In addition, variability in participants' prior experience with video games or gaming consoles may influence engagement and learning, introducing heterogeneity in intervention effects. Differences in teaching style and classroom environment may also affect the consistency and fidelity of intervention delivery. The relatively short duration of the intervention may further limit the ability to observe long-term changes in physical literacy or sustained physical activity behaviors.

Measurement-related constraints also warrant consideration. The study relies on self-reported measures for physical activity and gaming behavior, which are subject to recall bias and social desirability bias and may affect data accuracy. Moreover, participants' awareness of being observed during the intervention may lead to the Hawthorne effect, where behavior changes occur simply because of observation rather than the intervention itself. These factors should be considered when interpreting the findings. Performance-based assessments can carry measurement error and may not fully capture the breadth of PL in authentic contexts. Specifically, the study assessed selected perceptual-motor components (simple reaction time and eye-hand coordination) rather than comprehensive motor competence, which limits interpretation of PL development to specific attributes. The reliance on self-reported measures introduces potential biases that should be interpreted with caution. The tennis knowledge quiz used in this study was developed by the research team and pilot-tested for clarity and relevance. However, it has not been formally validated, which may introduce measurement bias and limit the generalizability of findings related to cognitive domain outcomes. Future studies should consider using or developing standardized, validated instruments to assess sport-specific knowledge.

Additionally, we acknowledge that the measures of cognitive and social engagement employed in this study are exploratory and context-specific. These constructs, such as attention, decision-making, peer interaction, and collaboration, are complex and multidimensional, and currently lack universally validated instruments for use in SVG interventions within PE settings. Our intention in including these measures was to provide preliminary insights into how students engage beyond physical outcomes, which we believe is critical for understanding the broader educational value of such interventions. However, we recognize that these findings should be interpreted with caution and may not be generalizable to other contexts. Future research should prioritize the development or adoption of standardized, validated tools that can reliably capture cognitive and social engagement across diverse populations and learning environments, and examine how these dimensions contribute to long-term educational and developmental outcomes.

## Theoretical contribution

This work aims to extend PL theory by exploring its application within technology-enhanced learning environments. Current frameworks primarily emphasize traditional pedagogical approaches, with limited consideration of digital tools. By incorporating SVG into structured PE lessons, this study seeks to demonstrate how interactive technologies can operationalize all four domains of PL: physical, cognitive, affective, and social, within a hybrid learning context. The inclusion of the social domain, assessed through the APLQ subset, addresses a notable gap in existing literature where social engagement is often underrepresented. These insights are expected to inform conceptual models that position digital interactivity as a legitimate pathway for physical literacy development.

## Practical implications

If successful, this intervention could provide educators with a novel strategy to enhance student engagement, motivation, and collaborative learning in PE settings. SVG-based activities may serve as an inclusive tool for students who are less confident in traditional physical activities or who respond positively to gamified experiences. The protocol also emphasizes alignment with curriculum objectives and classroom management considerations, which are critical for real-world implementation. Future research will be needed to evaluate long-term impacts on physical literacy and physical activity behaviors, as well as to refine best practices for integrating technology into PE curricula.

## Confidentiality

All data collected during the study will be handled with strict confidentiality. Informed consent will be obtained from all participants, and data will be anonymized using unique identifiers. The sample of the consent form is attached as Appendix 3. Personally identifiable information, such as names and contact details, will be stored separately from research data and excluded from all analyses and publications.

Data will be retained for five years, after which personal identifiers will be securely disposed of. Secure data collection methods—including encryption and restricted access protocols—will be employed to safeguard data integrity. Access to raw data will be limited to authorized members of the research team.

No data will be shared with third parties without explicit participant consent. Any data shared for academic or collaborative purposes will be fully anonymized. Participant confidentiality will be maintained throughout the research process and beyond.

## Data monitoring

The principal investigator (PI), along with designated research assistants, will oversee data monitoring throughout the study. Interim analyses will be conducted at predefined intervals to evaluate the efficacy and safety of the intervention. Any anomalies or unexpected patterns will be promptly investigated and reported to the PI. Based on the interim findings and other relevant considerations, the PI will determine whether to continue or terminate the trial. This decision-making process will follow established protocols and ethical guidelines [28].

## Patient and public involvement

Students were not involved in the initial design of this study. Both students and teachers will participate in the conduct, reporting, and dissemination phases. Teachers from the tennis classes involved in the study contributed to its design and will continue to be engaged through regular meetings to provide feedback and insights as the study progresses.

## Protocol amendments

Any modifications to the study protocol, including changes to eligibility criteria, outcomes, or interventions, will be communicated to the Survey and Behavioural Research Ethics Committee of the CUHK and to trial participants via email. These changes will be discussed and approved by the PI prior to implementation.

## Harms

Tennis class teachers involved in the study will be responsible for identifying, documenting, and reporting any adverse events or unintended effects related to the trial interventions. All incidents will be recorded in a pre-designed Excel file for systematic tracking and future reference. In the event of any such occurrences, appropriate healthcare professionals will be contacted immediately to provide assistance. Although the likelihood of musculoskeletal or other injuries from sports video gaming is low, minor adverse events such as discomfort, fatigue, or strain may still occur. In particular, physical testing procedures such as the Alternate Hand Ball Toss Test may pose a slight risk of upper limb strain or coordination-related discomfort due to repetitive movements. All minor adverse events will be categorized by type, severity, and relation to either the intervention or testing activity, and reviewed regularly by the research team to ensure participant safety and transparency throughout the study.

## Supporting information

**S1 File. SPIRIT fillable checklist.**
(DOC)

## Acknowledgments

The authors would like to thank the Physical Education Unit, The Chinese University of Hong for providing access to venues, equipment, and administrative support essential to the preparation of this study protocol. We also gratefully acknowledge the Faculty of Education, The Chinese University of Hong Kong for the institutional support.

## Author contributions

**Conceptualization:** Wai Keung Ho.

**Data curation:** Wai Keung Ho, Siu Ming Choi.

**Formal analysis:** Wai Keung Ho, Siu Ming Choi.

**Funding acquisition:** Wai Keung Ho, Raymond Kim Wai Sum.

**Methodology:** Wai Keung Ho, Raymond Kim Wai Sum, Siu Ming Choi.

**Project administration:** Wai Keung Ho.

**Resources:** Wai Keung Ho.

**Supervision:** Raymond Kim Wai Sum.

**Writing – original draft:** Wai Keung Ho.

**Writing – review & editing:** Wai Keung Ho, Raymond Kim Wai Sum, Siu Ming Choi.

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
