## [Decision Letter · Decision Letter 0]

20 Oct 2025

Dear Dr. Sum,

Thank you for submitting your manuscript to PLOS ONE. After careful consideration, we feel that it has merit but does not fully meet PLOS ONE’s publication criteria as it currently stands. Therefore, we invite you to submit a revised version of the manuscript that addresses the points raised during the review process.

We look forward to receiving your revised manuscript.

Kind regards,

Maheshkumar Baladaniya

Academic Editor

PLOS ONE

Journal Requirements:

**Additional Editor Comments:**

There are several points those need author's attention. Need to change manuscript accordingly.

Reviewers' comments:

Reviewer's Responses to Questions

**Comments to the Author**

1. Does the manuscript provide a valid rationale for the proposed study, with clearly identified and justified research questions?

Reviewer #1: Yes

Reviewer #2: Partly

2. Is the protocol technically sound and planned in a manner that will lead to a meaningful outcome and allow testing the stated hypotheses?

Reviewer #1: Yes

Reviewer #2: Partly

3. Is the methodology feasible and described in sufficient detail to allow the work to be replicable?

Reviewer #1: Yes

Reviewer #2: Yes

4. Have the authors described where all data underlying the findings will be made available when the study is complete?

Reviewer #1: Yes

Reviewer #2: Yes

5. Is the manuscript presented in an intelligible fashion and written in standard English?

Reviewer #1: Yes

Reviewer #2: Yes

You may also provide optional suggestions and comments to authors that they might find helpful in planning their study.

Reviewer #1: Summary:

The study addresses an important gap in the literature, particularly in Asian university populations, and integrates both objective and self-reported measures across multiple domains of physical literacy. The protocol is thorough, with clear inclusion/exclusion criteria, a detailed intervention plan, and ethical considerations.

Comments

The introduction is dense with literature but could benefit from clearer linkage between gaps in knowledge and the study’s specific aims.

While the sample size calculation is thorough, there is slight ambiguity regarding whether the intra-cluster correlation (ICC) was considered. Cluster designs require adjustments for ICC to avoid underestimating sample size.

Two 20-minute sessions per week may be relatively short to elicit measurable changes in physical literacy, especially in physical competence. Some rationale for the chosen duration could strengthen the protocol.

It may also be useful to clarify how variability in participant adherence (e.g., first-come-first-served access to consoles) will be handled analytically.

The tennis knowledge quiz is researcher-designed and pilot-tested but not formally validated. Consider reporting limitations and potential biases in interpretation.

Clarification on handling potential confounders, covariates, and interaction effects would strengthen the analysis plan.

The discussion section currently highlights the impact of sports video games on physical activity and literacy but lacks explicit linkage to evidence demonstrating the broader psychological and physiological benefits of structured physical activity interventions. Add below phrase to strengthen the practical relevance and theoretical grounding of the findings.

“Previous work has demonstrated that structured physical activity interventions can positively influence psychological and physiological outcomes, including mood regulation, emotional resilience, and neurochemical modulation [Nasif et al., 2025 https://doi.org/10.31579/2578-8868/359 ]. These findings underscore the broader role of physical engagement through innovative modalities such as sports video games to promote physical literacy, enhance motivation, and support overall health among university students.”

While harms are mentioned, the likelihood of musculoskeletal or other injuries from video gaming is probably low; however, the protocol could specify how minor adverse events will be reported and categorized.

The protocol does not describe whether students in the control group are discouraged or restricted from gaming independently. This may introduce contamination bias.

Limitations Section: Acknowledge the reliance on self-reported measures and potential biases. Note that cognitive and social engagement measures are largely exploratory and context-specific.

Reviewer #2: In abstract there is no explicit research question or hypothesis mentioned upon which papers usually based. It can be seen that more focus on procedural detail is given (e.g., “two 20-minute sessions per week for six weeks”); this level of detail belongs in Methods, not the abstract. One more thing can be seen as more focus is put on general background (sales data, gaming popularity) rather than specific causal mechanisms. There are some repetition of phrases present such as “motivational and cognitive aspects,” “complementing PE lessons,” which dilute precision.

There is one thing that can be observed which is lack of explanation of why certain PL measures (e.g., APLQ subset) were selected over full validated scales. There are no mention of limitations or risk nor any discussion took place in the manuscript. (e.g., Hawthorne effect, self-report bias). Also, lacks “theoretical contribution” narrative—merely describes procedures present that can be improved upon.

Author can also take example such as - “AI-assisted interactive systems have demonstrated increased engagement and comprehension in technology-mediated environments, supporting the educational potential of game-based interventions.” DOI - doi.org/10.47363/JAICC/2024(3)380

Authors can explain and expand cultural adaptation steps (translation, back-translation, pilot testing). There is also scope of including mixed-model analysis (random effects for clusters), report partial eta-squared and confidence intervals for primary outcomes.

**Do you want your identity to be public for this peer review?** For information about this choice, including consent withdrawal, please see our Privacy Policy

Reviewer #1: No

Reviewer #2: No

---

## [Author Response · Author response to Decision Letter 1]

12 Nov 2025

We thank the editor and reviewers for their constructive comments. Please refer to the attached document titled “Response to Reviewers” for details on the points we have amended.

---

## [Decision Letter · Decision Letter 1]

4 Jan 2026

Assessing the impact of a tennis-themed sports video game on physical literacy and participation on physical activities among university students: a cluster randomized controlled trial study protocol

PLOS One

Dear Dr. Sum,

Thank you for submitting your manuscript to PLOS ONE. After careful consideration, we feel that it has merit but does not fully meet PLOS ONE’s publication criteria as it currently stands. Therefore, we invite you to submit a revised version of the manuscript that addresses the points raised during the review process.

https://journals.plos.org/plosone/s/submission-guidelines#loc-laboratory-protocols . Additionally, PLOS ONE offers an option for publishing peer-reviewed Lab Protocol articles, which describe protocols hosted on protocols.io. Read more information on sharing protocols at https://plos.org/protocols?utm_medium=editorial-email&utm_source=authorletters&utm_campaign=protocols .

We look forward to receiving your revised manuscript.

Kind regards,

Maheshkumar Baladaniya

Academic Editor

PLOS One

**Journal Requirements:**

**Additional Editor Comments:**

There are some minor changes required which have been raised by the reviewer. After that manuscript is acceptable to publish.

Reviewers' comments:

Reviewer's Responses to Questions

**Comments to the Author**

1. Does the manuscript provide a valid rationale for the proposed study, with clearly identified and justified research questions?

Reviewer #2: Partly

Reviewer #3: Yes

2. Is the protocol technically sound and planned in a manner that will lead to a meaningful outcome and allow testing the stated hypotheses?

Reviewer #2: Yes

Reviewer #3: Yes

3. Is the methodology feasible and described in sufficient detail to allow the work to be replicable?

Reviewer #2: Yes

Reviewer #3: Yes

4. Have the authors described where all data underlying the findings will be made available when the study is complete?

Reviewer #2: Yes

Reviewer #3: Yes

5. Is the manuscript presented in an intelligible fashion and written in standard English?

Reviewer #2: Yes

Reviewer #3: Yes

You may also provide optional suggestions and comments to authors that they might find helpful in planning their study.

Reviewer #2: Abstract subtly overextends its claims through implication rather than direct assertion. The inclusion of objective measures such as reaction time and eye–hand coordination suggests physical competence enhancement, yet the abstract does not explicitly articulate how a sedentary gaming intervention is expected to influence these outcomes.

Introduction adopts a relatively uncritical stance toward the physical literacy construct itself. While the authors accurately describe PL as multidimensional, they do not sufficiently acknowledge ongoing theoretical debates regarding its measurement and operationalization.

The hypotheses related to physical competence warrant closer scrutiny. The manuscript treats improvements in reaction time and eye–hand coordination as indicators of enhanced physical competence attributable to the intervention. This framing risks conflating perceptual–cognitive improvements with embodied motor competence.

Although the manuscript notes that courses follow standardized content, variability in teaching style, enthusiasm, and informal reinforcement of the intervention is inevitable in real-world PE settings. While this does not invalidate the design, it should be more explicitly acknowledged and analytically addressed.

What is less developed is the explicit articulation of the intervention’s theoretical mechanism. While the manuscript describes what participants do during gameplay, it stops short of clearly mapping specific game mechanics to expected cognitive, affective, or behavioral outcomes.

Reviewer #3: Authors have done well job on revising their manuscript. I think manuscript has improved in a great level.

**Do you want your identity to be public for this peer review?** For information about this choice, including consent withdrawal, please see our Privacy Policy

Reviewer #2: No

Reviewer #3: No

---

## [Author Response · Author response to Decision Letter 2]

6 Jan 2026

Please refer to the attachment 'Response to Reviewers'.

---

## [Decision Letter · Decision Letter 2]

10 Feb 2026

Assessing the impact of a tennis-themed sports video game on physical literacy and participation on physical activities among university students: a cluster randomized controlled trial study protocol

PONE-D-25-48676R2

Dear Dr. Sum,

We’re pleased to inform you that your manuscript has been judged scientifically suitable for publication and will be formally accepted for publication once it meets all outstanding technical requirements.

Kind regards,

Maheshkumar Baladaniya

Academic Editor

PLOS One

Additional Editor Comments (optional):

Manuscript is acceptable for the publication.

Reviewers' comments:

Reviewer's Responses to Questions

**Comments to the Author**

1. Does the manuscript provide a valid rationale for the proposed study, with clearly identified and justified research questions?

Reviewer #2: Partly

Reviewer #3: Yes

2. Is the protocol technically sound and planned in a manner that will lead to a meaningful outcome and allow testing the stated hypotheses?

Reviewer #2: Partly

Reviewer #3: Yes

3. Is the methodology feasible and described in sufficient detail to allow the work to be replicable?

Reviewer #2: Yes

Reviewer #3: Yes

4. Have the authors described where all data underlying the findings will be made available when the study is complete?

Reviewer #2: Yes

Reviewer #3: Yes

5. Is the manuscript presented in an intelligible fashion and written in standard English?

Reviewer #2: Yes

Reviewer #3: Yes

You may also provide optional suggestions and comments to authors that they might find helpful in planning their study.

Reviewer #2: The cluster randomized controlled trial design is appropriate and well justified for an educational context. Randomization at the class level minimizes contamination and respects institutional constraints. The stratification by gender is also sensible.

The Introduction is well written and well referenced, but it is doing too much conceptual work without sufficient narrowing. In particular, the manuscript positions physical literacy (PL) as a holistic, embodied construct while simultaneously proposing a sedentary sports video game (SVG) intervention.

The intervention is described in strong logistical detail (SVG Corner setup, scheduling, attendance tracking), which is commendable and often missing from protocols.

The use of both objective and self-reported measures is appropriate and well justified. Standardized tests (RDT, AHWT) enhance replicability, and validated questionnaires (IPAQ-SF, PPLI, SMS-6) are used appropriately.

The planned use of mixed-effects models is a clear strength and appropriate for clustered longitudinal data. Adjusting for baseline covariates and exploring subgroup effects demonstrates analytic sophistication.

Reviewer #3: Authors have done well job on revising their manuscript. Manuscript is ready to be published. Thank you!

**Do you want your identity to be public for this peer review?** For information about this choice, including consent withdrawal, please see our Privacy Policy

Reviewer #2: No

Reviewer #3: No

---

## [Editor Report · Acceptance letter]

PONE-D-25-48676R2

PLOS One

Dear Dr. Sum,

I'm pleased to inform you that your manuscript has been deemed suitable for publication in PLOS One. Congratulations! Your manuscript is now being handed over to our production team.

Kind regards,

on behalf of

Dr. Maheshkumar Baladaniya

Academic Editor

PLOS One